# Extraction Solvents Affect Anthocyanin Yield, Color, and Profile of Strawberries

**DOI:** 10.3390/plants12091833

**Published:** 2023-04-29

**Authors:** Toktam Taghavi, Hiral Patel, Reza Rafie

**Affiliations:** 1Agricultural Research Station, Virginia State University, Petersburg, VA 23806, USA; patelhiral712@gmail.com; 2Cooperative Extension, Virginia State University, Petersburg, VA 23806, USA; reza.rafie60@gmail.com

**Keywords:** flavonoids, anthocyanidins, extraction solvent, organic solvent, methanol, ethanol, chloroform, pigments

## Abstract

Anthocyanins are a major group of plant pigments that have antioxidant activities. Pigments play a major role in human health and have attracted a lot of attention globally. Many factors affect anthocyanin yields, such as solvent type, incubation time, solvent-to-sample ratio, sample type, and temperature. The first parameter was tested, and the rest were considered constant in this experiment. A total of nine organic and water-based solvents (methanol and chloroform: methanol, acetone, ethanol, water) and their combinations were compared to extract anthocyanins from freshly-pureed strawberries. Solvents changed anthocyanin yield, color parameters, and profile. The color parameters of a* values lower than 30, L* values higher than 85, hue angle more than 40, and chroma less than 30 indicated some color degradation in strawberry anthocyanins. Therefore, the best solvents for anthocyanin assessment were methanol and methanol: water. The second-best solvent was the pH differential buffers. Other solvents such as ethanol, chloroform: methanol, water, and water-based solvents extracted considerable amounts of anthocyanins; however, they showed some degree of color degradation, evidenced by the color parameters. Acetone did not yield a stable extract which degraded over 48 h of storage at 4 °C. The extraction solvent determined the main anthocyanin of the anthocyanins profile. Pelargonidin was the major anthocyanin in chloroform: methanol solvent, while delphinidin was dominant in all other solvents.

## 1. Introduction

Fresh fruits like strawberries are rich in antioxidants such as anthocyanins [1]. Research on strawberry anthocyanins identification and physiological function has attracted a lot of attention [2,3]. Therefore, identifying the best extraction and quantitative analysis methods for different strawberry samples is desired [3].

The spectroscopic method is an inexpensive quantitative analysis method that determines total anthocyanin concentration using solvents. Quantitative analytical methods such as HPLC, although expensive, accurately identify the extracted anthocyanin profile [4]. The ultrasonic-assisted method is another method that produces energy using ultrasonic. Most often, the energy is not homogeneously distributed, reducing the efficiency of extraction. However, the solvent extraction method can preserve the antioxidant properties of anthocyanins to a greater extent than the ultrasonic method [5]. Therefore, the solvent extraction method was used for this experiment because it is an accurate, simple, and rapid method of extracting and measuring monomeric anthocyanin content with high yield. The HPLC method was used to identify the anthocyanins profile.

The two main solvents used for anthocyanin extraction are organic solvents (i.e., methanol, ethanol, or acetone) and water-based solvents (i.e., pH differential buffers [6]). Both extract anthocyanins in an acidic environment [7,8] and are subject to interference from light-absorbing impurities present in the extracts [6].

Solovchenko et al. [6] claimed chloroform should be used to remove light-absorbing impurities (i.e., lipids) for anthocyanin assessment in apple peels. Neff and Chory [9] also used chloroform: methanol in a different ratio for Arabidopsis leaf samples. Chloroform also dissolves some anthocyanins, such as pelargonidin [10].

Methanol, ethanol, and acetone have been used as solvents for extracting anthocyanins from strawberries, with the first being the most commonly used, along with pH differential buffers [11]. Karaaslana and Yaman [12] have shown that the anthocyanin contents of strawberry fruit change by altering extraction solvents. The best extraction solvent in this fruit was methanol [12].

The pH differential buffer creates reversible structural changes in anthocyanins at different samples’ pH [13]. At a pH of 1.0, the molecule strongly absorbs light at 520 nm, and at a pH of 4.5, is colorless [14]. Thus, the pigment concentration is proportional to the difference in absorbance at 520 nm [11], and accurately estimates the total anthocyanins [15]. However, the extracted material is often hazy, and the impurities interrupt the spectroscopic process. Combined solvents, suggested by Gauch et al. [16], extract the anthocyanin with acidified methanol and then calculate the anthocyanin concentration with the pH differential buffer [3,16]. The benefit of the combined solvents method is that it removes haze produced in pH differential buffers, as the haze makes the pH differential method vulnerable [3].

The anthocyanin extracts may undergo hydrolysis of the anthocyanin (i.e., when concentrated by vacuum rotary evaporation) or deacylation of anthocyanins acylated with aliphatic acids. To obtain anthocyanins closer to their natural state, acetone was suggested by Garcia-Viguera et al. [17] for strawberries for their pectin clotting properties. According to Garcia-Viguera et al. [17], using methanol gives rise to difficulties concerning sample concentration and filtration for further high-performance liquid chromatographic analysis. They suggested that using acetone allows an efficient and more reproducible extraction, avoids problems with pectins, and permits a much lower temperature for sample concentration. The method was applied to analyze anthocyanins from four varieties of strawberries (‘Camarosa’, ‘Oso Grande’, ‘Chandler’ and ‘Tudla’; [17]).

There is a report that the major anthocyanins of strawberries are cyanidin-3-glucoside, pelargonidin-3-glucoside, 3-rutinoside, and 3-malonyl glucoside [18]. DZhanfezova et al. [19] also mentioned that pelargonidin is present in higher concentrations in modern cultivars due to selections for brighter red colors, expressed by pelargonidin, compared to cyanidin. Da-Silva et al. [20] identified pelargonidin as the main anthocyanin in five strawberry cultivars. The differences could be due to the cultivar or other extraction variables [19].

Color is one of the most critical properties of fruits and vegetables and is the basis of their acceptability and preference. Fruit colors can sometimes represent the degree of ripeness and the fruit quality. Naturally-derived colorants are a viable alternative to synthetic ones and are in high demand. However, the availability of natural pigment sources, extraction, and stability of colorants must be considered in producing colorants. Anthocyanins are natural colorants that are extracted from a wide variety of sources, such as fruits and flowers [21].

Anthocyanins are water-soluble and usually found in an aqueous environment in nature. The study of colorants cannot leave the solvent and its interaction with the solute out of the picture. Because anthocyanins express their chromatic function in solution, their color strongly depends on the environment, the concentration, the acidity, the polarity, and the temperature of the solvent or the presence of otherwise optically inert substances in solution, which alter its color [22].

Our preliminary experiments have shown that different solvents create different anthocyanin profiles and colors. Therefore, in this experiment, we compared different organic and water-based solvents (methanol and chloroform: methanol with and without water, water, acetone, ethanol, and pH differential buffers) for fresh strawberry puree. These solvents were used because they have been the main solvents mentioned in the literature to extract anthocyanins from fresh fruits, including strawberries. This experiment had two objectives. The first was to study the anthocyanin yield and color of the extracts from different solvents using the CIELAB parameters. The second was to study the anthocyanin profile of the extracted strawberries.

## 2. Results

### 2.1. Total Anthocyanins

All solvents used freshly pureed strawberries from a homogenized bulk sample and had the same dilution factor (solvent: sample ratio); therefore, differences in anthocyanin concentrations, profile, and color were related to the ability of the specific solvent to soak and extract anthocyanins, not sample differences or the solvent-to-sample ratio.

Among the solvents tested, chloroform: methanol (by Solovchenko et al. [6]), followed by ethanol [12], and methanol [12] extracted the most amounts of anthocyanins (~10–12 A/gFW). Solvents that had methanol: water (Neff and Chory [9]; and Lindo & Caldwell [7], pH differential buffers [11], and acetone [17]) extracted less anthocyanins (6.8–8.8 A/gFW) with acetone extracts being unstable and losing color within 48h. The water-based solvents (combined solvents of Gauch et al. [16]) and water [17] were the least effective solvents (<5 A/gFW) in extracting anthocyanins from the strawberry puree (Table 1).

### 2.2. Transmittance Spectrum

The extracts with higher anthocyanin concentration had lower transmittance. In the transmittance spectrum measured by CM-5, solvents with methanol, such as those used in Solovchenko et al. [6], Lindoo and Caldwell [7], and Neff and Chory [9] had the same spectrum pattern and transferred the least amount of light. Although chloroform was used in the Neff and Chory [9] buffer, it separated during the centrifugation process due to water availability in the buffer; therefore, it did not play a role in the anthocyanin color assessment. These methanol extracts had the lowest transmittance at λ_min_ of 520 nm (Figure 1).

The extracts by water-based solvents such as pH differential buffers [11], water, combined solvents [16], and ethanol transmitted more light than methanol-based solvents. However, the minimum transmittance wavelength (λ_min_) was 500 nm for water-based solvents and 520 nm for ethanol. The pH differential method is based on the differences between the transmittance of the colorless structure of anthocyanins at pH 4.5 and the pigmented anthocyanins at pH 1.0. Therefore, the difference reflects monomeric anthocyanins with a minimum absorbance (λ_min_) of around 500 nm.

The chloroform: methanol solvent (by Solovchenko et al. [6]) had the highest transmittance of all the pigmented extracts with the λ_min_ at 520 nm. Acetone extract lost its color during the 48 h of storage at 4 °C and showed maximum transmittance within the 400–700 nm range (Figure 1).

### 2.3. Color Parameters

The evaluation of the color is based on the use of the CIELAB color system (Commission Internationale de l’Eclairage CIE, 1986) in which the L*, a*, and b* values were measured by spectrophotometry (transmittance) and described in a three-dimensional color space, where there are three axes for each color. Axis L* represents the lightness, from completely opaque (0) to completely transparent (100); a* represents redness (or –a* of greenness), and b* represents yellowness (or –b* of blueness) with values ranging from −128 to +127. Chroma gives further information on the saturation or intensity of color. A confounding phenomenon regarding chroma is that it will increase with pigment concentration to a maximum and then decreases as the color darkens [24]. In the high ratio of solvent to sample (20:1) used in this experiment with the low concentration of the pigments, chroma should increase with pigment concentration. In this experiment, solvents significantly changed the extracted anthocyanins’ color parameters at *p* ≤ 0.01 (Table 2). The differences in color parameters may be affected by differences in the pH of the extracts as chloroform: methanol solvent [6] had a pH of 3.0. In contrast, other solvents had a pH between 1.0–2.0.

The L* and a* values appeared to better explain the variations of colors created by different solvents (Figure 2A). Both L* and a* divided the samples into different groups. The first group is the methanol and methanol: water solvents (used by Neff and Chory [9], Lindoo and Caldwell [7], and Solovchenko et al. [6]) with the lowest L* and highest a* values with minimal variations and an average of 81 and 34 for L* and a*, respectively, which created a rose color (Table 2; Sundberg [26]).

The a* values decreased, and the L* increased, in other solvents in this order: pH differential buffers [11], Ethanol [12], chloroform: methanol [6], and combined solvents [16]. In the second step, the combined solvents [16] used pH differential buffers (water-based) to assess anthocyanins; therefore, the color values were close to the water solvent. The acidified water was the next group in increasing L* and decreasing a* values, and this created a peach color (Table 2 and Figure 2A).

The extracts were stored at 4 °C for 48 h before being analyzed by Spectrophotometer CM-5 to observe if they were stable after extraction. The color of acetone extracts was unstable over the storage period, and faded away (Table 2). Therefore, acetone is in a separate group very far from other solvents with very low a* and b* values and high L* values (Figure 2A).

Hue angles were lowest when chloroform: methanol [6] and ethanol [12] were used as solvents. The methanol and methanol: water solvents [6,7,9] had a high hue angle with an average of 21.2. The hue angle increased significantly when the water [12] and water-based solvents (pH differential [11]) and combined solvents [16] were used. Acetone had the highest hue angle primarily due to the complete degradation of anthocyanins after storage for 48 h (Figure 2B). Higher hue angle means more yellow color, indicating more degradation of anthocyanins. The higher hue angles of the acetone, combined solvents, and water-based solvents (water and pH differential) confirm that the anthocyanins were fully or partially degraded using these solvents.

Chroma was highest in methanol and methanol: water solvents [6,7,9] with an average of 36.5 (Figure 2B). Chroma decreased to 31.8 in pH differential buffers [11] and decreased to less than 25 in other solvents, (ethanol, combined solvents, water, chloroform: methanol and acetone). In the high ratio of solvent to sample (20:1) used in this experiment with the low concentration of the pigments, chroma should increase with pigment concentration. The decreased chroma in these solvents confirmed the reduced pigment concentration.

### 2.4. UHPLC Profiling of Individual Anthocyanins

Extracted samples were stored for 48 h before being analyzed for anthocyanin identification using the UHPLC-MS system. The profile was compared with 103 anthocyanin standards. Fifty anthocyanins (out of 103) were detected in the samples analyzed by the UHPLC-MS system (Table 3) and the chromatograms presented in Appendix A. The relative retention times (RT) were detected at the expected retention times [27].

The two main anthocyanins were delphinidin (C_15_H_11_O_7_) and pelargonidin (C_15_H_11_O_5_). Methanol-based solvents [6,7,9] and pH differential buffers [11] extracted delphinidin as the main anthocyanin. The methanol: chloroform solvent extracted a more diverse anthocyanin profile from samples. The main anthocyanin was pelargonidin, and other major anthocyanins were cyanidin and petunidin. However, in all other solvents, delphinidin was the dominant anthocyanin, compromising about 40–87% of anthocyanins. Depending on the solvent used, cyanidin (C_15_H_11_O_6_), petunidin (C_16_H_13_O_7_), and Delphinidin-3,5-o-diglucoside (C_27_H_31_O_17_) were also present in smaller quantities.

Different solvents extracted different anthocyanin profiles. The pH of the solvents may be responsible for the differences in the profile, as the chloroform: methanol solvent [6] had a pH of 3.0 while other solvents had a pH between 1.0 and 2.0.

## 3. Discussion

### 3.1. Total Anthocyanins

Many factors affect anthocyanin yields, such as solvents, incubation time, solvent: sample ratio, sample type, and temperature. The first parameter was tested, and the rest were considered constant in this experiment. All solvents were incubated for 24 h uniformly, except pH differential buffers and acetone solvents, which had a 20 min incubation time. Based on our previous report, the 20:1 (v/m) was the best ratio to keep the transmittance/absorbance in the linear range of the spectrophotometer [28].

Anthocyanins are highly soluble in water; however, anthocyanins dissolve in organic solvents such as methanol and ethanol due to anthocyanins’ polyphenolic structure, which adds a hydrophobic characteristic to them. Very few anthocyanins, such as pelargonidin, are also soluble in chloroform, and its addition to the buffer may increase total anthocyanin content [29]. The higher amount of anthocyanins in chloroform: methanol solvent [6] could be due to the addition of chloroform as a result of the increased solubility of specific anthocyanins, such as pelargonidin, compared to other solvents that lack chloroform. Adding both solvents (chloroform and methanol) will dissolve a broader range of anthocyanins than each solvent alone [10]. The addition of chloroform may affect the anthocyanin yield and determines what extraction solvent is superior in the presence or lack of it. Chloroform increased the anthocyanin yield as was seen in the chloroform: methanol solvent used by Solovchenko et al. [6]. A reason for higher anthocyanin yield in the presence of chloroform is the solubility of pelargonidin, of which strawberry is a rich source [3].

The solvent used in Neff and Chory [9] had methanol, water, and chloroform in the buffer; however, chloroform separated during the centrifugation process from water and did not play a role in the assessment of the anthocyanins, their color parameters, and profile. The anthocyanin concentration extracted from methanol: water was in the same range reported by other researchers [2]. The low anthocyanin content of the pH differential buffers was also seen in frozen strawberry samples [28].

Unlike Garcia-Viguera et al. [17], where they reported that the acetone solvent was superior to the ones containing methanol for extracting total phenolic compounds, we observed that although anthocyanin concentration was comparable to methanol, they degraded during storage and anthocyanin color vanished within 48 h at 4 °C.

According to Karaaslan and Yaman [12], solvent type is an essential factor in the extraction procedure. It affected anthocyanin yield, and strawberry samples extracted with acidified methanol had higher anthocyanins than acidified water, acetone, acetonitrile, and ethanol. Similarly, in this experiment, acidified methanol extracted a high amount of stable anthocyanins during storage at 4 °C.

### 3.2. Color Parameters

Visible color is not directly related to pigment concentration, and the human eye lacks linear sensitivity (Martinsen et al. [30]). Therefore, measuring color parameters by colorimetric methods is a more accurate way of measuring degradation in anthocyanin colors. Additionally, the color degradation is not visually noticeable, and needs analytical tools such as a colorimeter to be identified.

Patil et al. [21] stated that the hue angle and chroma values of alcoholic extracts from red radish skin were less when compared to acidified water, mainly due to the change in pH of the extracts. The pH values of the alcoholic extracts were averaged at 6.0, while acidified water had a pH of 1.7. They stated that the highest hue and chroma angle wer obtained at 50% ethanol in acidified water and 2% acidity. In our experiment, HCl (0.1%) was added to all solvents, and the pH was in the range of 1.0–2.0 for all solvents, except the chloroform: methanol solvent [6], where the pH was 3.0. The changes in color parameters due to the higher pH of chloroform: methanol were probable. The addition of chloroform may have been responsible for the change in color of the extracted anthocyanins.

An increase in the hue angle indicates color degradation, since the blue/purple color is associated with anthocyanin content and antioxidant activity [24]. Another critical measure is chroma, which indicates color saturation. In this experiment’s high solvent: sample ratio (20:1), chroma was expected to increase with pigment concentration.

An increase in hue angle and less intensity in chromaticity has been reported after the thermal processing of strawberry jam, suggesting the destruction of anthocyanins due to higher processing temperatures [30].

Similar results showed the degradation of blue corn chip anthocyanins due to higher processing temperatures and loss of their characteristic color, which induces marked structural changes in addition to the loss of their characteristic red-purple color [24].

Improved color and anthocyanin retention was reported by excluding oxygen during strawberry puree preparation. Howard et al. [31] reported lower L* and hue, and higher chroma values due to greater color stability when excluding oxygen.

Garzon and Wrolstad [32] reported pigment degradation in fortified strawberry juices with pelargonidin when the hue angle increased (from 40 to 75) and chroma decreased (from 30 to 15) during storage. They also reported increased hue angle and a considerable drop in chroma after storing strawberry concentrate.

According to the literature, a hue angle of more than 40 and chroma of less than 30 indicate some level of degradation in the anthocyanins of strawberries [32]. In this experiment, the hue angle was in the optimum range (less than 40) in all solvents except acetone. The chroma was less than 30 in ethanol, chloroform: methanol, water, acetone, and combined solvents showing partial degradation of anthocyanins in these solvents. However, the chroma was higher than 30 in methanol and methanol: water solvents [6,7,9], indicating the highest stability of anthocyanin colors in methanol and methanol: water solvents.

### 3.3. UHPLC Profiling of Individual Anthocyanins

Natural colorants are getting a lot of attention in the food industry and similar applications. Consumers are interested in using alternatives to synthetic colorants and look forward to using natural colorants for safety concerns and for their health benefits. Plant breeders are following these trends and looking for ways to integrate easy and quick pigment extraction protocols in their breeding programs so that they can assess the pigments of their breeding material effortlessly. They aim to create varieties with higher bioactive compounds and specific colors in flowers, leaves, and fruits [19].

Several reports have identified the main anthocyanins in strawberries; however, they all used different solvents, techniques, and varieties [28]. At least 25 anthocyanins have been identified in strawberry cultivars and selections. In a report on 18 strawberry cultivars or selections, Dzhanfezova et al. [19] identified pelargonidin 3-O-glucoside as the main anthocyanin. Da Silva [20] presented pelargonidin 3-O-glucoside, pelargonidin 3-O-rutinoside, and cyanidin 3-O-glucoside as the three main anthocyanins, representing more than 95% of strawberry anthocyanins.

Donno et al. [33] identified four major anthocyanins: cyanidin-3-glucoside, pelargonidin-3-glucoside, pelargonidin-3-rutinoside, and pelargonidin-3-acetylglucoside in five strawberry selections by using HPLC-DAD/MS. Karaaslan & Yaman [12] determined that the dominant anthocyanin for strawberries was pelargonidin-3-o-glucoside. Kawanobu et al. [34] reported that the major anthocyanins in the strawberry cultivars they studied were cyanidin-3-glucoside, pelargonidin-3-glucoside, and pelargonidin-3-malonylglucoside [12].

Pelargonidin-3-O-glucoside was about ten times higher than cyanidin-3-o-glucoside, and the content of delphinidin-3-o-glucoside and malvidin-3-o-glucoside was less than the detection limit in HPLC-ESI (electrospray ionization-MS [12]).

Kelebek and Selli [35] determined the anthocyanins of three strawberry cultivars to be cyanidin-3-glucoside, cyanidin-3-rutinoside, pelargonidin-3-glucoside, pelargonidin-3-rutinoside, pelargonidin-3-malonyl-glucoside, and pelargonidin-3-acetyl-glucoside. Goiffon et al. [36] reported pelargonidin- 3-arabinoside as the third anthocyanin in two varieties (‘Senga sengana’ and ‘El Santa’) instead of pelargonidin-3-rutinosied. According to Tamura et al. [37], Pelargonidin-3-malonylglucoside is one of the main anthocyanins in strawberries.

In general, pelargonidin 3-O-glucoside provides a bright red color to strawberries, whereas cyanidin 3-O-glucoside provides a darker red color [32]. Thus, the consumer preference over time for a bright red color has indirectly led to the selection of strawberry cultivars with pelargonidin 3-O-glucoside as the major anthocyanidin form, representing 70–90% of the total anthocyanins regardless of genetic and environmental factors [3].

Neder-Suárez et al. [24] expressed that extraction methods significantly affect Cyanidin-3-glucoside content and could similarly affect other anthocyanins. Ghassempour et al. [27] also discovered that different extraction methods (organic solvent, ultrasound, and microwave-assisted) extracted different anthocyanin profiles from grape skin in their experiment. Malvidin-3-glucoside was the major compound from the organic solvent method, whereas in the ultrasound and microwave-assisted methods, other compounds such as malvidin-(6-coumaroyl)-3-glucoside, petunidin-(6-coumaroyl)-3-glucoside, and malvidin-(6-caffeoyl)-3-glucoside were dominant. We also identified different anthocyanins as the major compound in different extraction solvents. Delphinidin was the main anthocyanin in all solvent types except chloroform. The addition of chloroform has changed the profile of the extracted anthocyanins. Pelargonidin was the major compound in the chloroform: methanol solvent (Solovchenko et al. [6], pH 3.0). Similarly, da Silva et al. [20] identified pelargonidin as the main anthocyanin in most strawberry cultivars tested. The observed behavior resulted from different extracting mechanisms or pH involved in the processes, as suggested by Ghassempour et al. [27]. Similarly, in this experiment, the pH was 1.0–2.0 for all solvents, except the chloroform: methanol solvent [6], where the pH was 3.0.

## 4. Materials and Methods

### 4.1. Sample Preparation

Strawberry cultivar Allstar was grown under a high tunnel at the Randolph Research Farm of Virginia State University, Petersburg, VA, USA (37°2′ N, 77°4′ W) during 2018–2019. The fruits were harvested from the open field on 1st June 2021, and only firm fruits with uniform size and red color with no blemishes were selected for the experiment. About 300 g of strawberry fruits were pureed in a blender (Magic Bullet 600-Watt) to create a bulk representative sample (fresh puree). The bulk homogenization eliminated the inherent sample differences. One gram of the puree was weighed in a 50 mL centrifuge tube with closed lids and was subjected to extraction with different solvents.

### 4.2. Total Anthocyanins Extraction

A total of nine extraction solvents (or combinations) were compared to extract anthocyanins from freshly pureed samples. All solvents were acidified with 0.1% HCl, except in methanol: water (1% HCl; [7]) and pH differential and combined solvents, where HCl was added to the pH buffers to reach a certain pH (1.0, and 4.5).

As the first solvent, twenty ml of acidified methanol (0.1% HCl) was added to the strawberry samples, according to Solovchenko et al. [6]. For the second solvent, the methanol: water solvent suggested by Neff and Chory [9] was used to extract anthocyanins. For this solvent, 15 mL methanol, 10 mL water (methanol: water), and 0.15 mL HCL were mixed as the buffer. Twenty ml of the buffer and 20 mL of chloroform were mixed thoroughly with the samples. Although chloroform was added to this mixture, it separated during centrifugation and was not present during assessments.

For the third solvent, the samples were mixed with 20 mL of methanol: water: HCl (80:20:1; [7]). As the fourth solvent, 20 mL of acidified ethanol (0.1% HCl) was mixed with the samples (Karaaslan & Yaman [12]).

The fifth solvent was suggested by Solovchenko et al. [6] by adding chloroform to the mixture. Anthocyanins were extracted by adding 20 mL of chloroform: methanol (2:1 *v*/*v*, acidified with 0.1% HCl) to the samples; however, chloroform did not separate during centrifugation.

For the sixth solvent, acidified acetone (0.1% HCl) was mixed with the samples in the ratio of 20 mL solvent: 1 g sample. To test the seventh solvent, 20 mL of acidified water (0.1% HCl) was added to the samples in the ratio of 20 mL solvent: 1 g sample [17].

The mixtures obtained from solvents one to seven were thoroughly mixed and incubated at 4 °C (precision refrigerated incubator, Thermo Fisher Scientific, Waltham, MA, USA) in the dark on a shaker (Thermo Fisher Scientific, Waltham, MA, USA) for 24 h [12]. The homogenates were centrifuged (Heraeus Multifuge X1R Benchtop Refrigerated Centrifuge, Thermo Fisher Scientific, Waltham, MA, USA) at 4 °C, 7000 rpm for 15 min. The supernatant was removed, and the spectrophotometer measured the absorbance (Genesys 150 UV-Vis connected to Visionlite 5 software, Thermo Fisher Scientific, Waltham, MA, USA) at 530 and 657 nm [7,23]. Anthocyanin concentrations in the extracts were determined by using Formula (1). They were given as A/g fresh fruit tissue [3], where TA = total anthocyanin, A = absorbance at 530 and 657 nm, V = volume of extract (ml), and M = fresh mass of the sample (g).
(1)$$\;\mathrm{TA}\;=\frac{\left(A_{530}-0.3{\;A}_{657}\right)\times V}{M}$$

The eighth solvent was the pH differential buffers (pH 1.0 and 4.5; [11]) used to calculate strawberry fruit anthocyanin content. The strawberry samples were mixed thoroughly with 20 mL of either buffer pH 1.0 (0.025 M potassium chloride) or buffer pH 4.5 (0.4 M sodium acetate buffer) and then incubated for 20 min at room temperature and centrifuged at 4 °C, 7000 rpm for 15 min. The supernatant was then removed, the absorbance was read at 520 and 700 nm, and anthocyanin was calculated by using Formula (2):(2)$$\mathrm{TA}=\frac{A\times V}{M}$$
where A = (A520 nm − A700 nm) pH 1.0 − (A520 nm − A700 nm) pH 4.5; V = volume of extract (ml), and M = fresh mass of the sample (g).

In a two-step process, methanol: water: HCl and pH buffers were used as the ninth solvent (combined solvents) as suggested by Gauch et al. [16]. In the first step, 20 mL of methanol: water: HCl buffer (80:20:1) was added to the strawberry samples and incubated in darkness for 24 h. The crude extract obtained was centrifuged at 4 °C and 7000 rpm for 15 min. The supernatant was removed (step 1), and half of the original volume (10 mL) was concentrated under vacuum (−10 Evaccume psi) at 35 °C overnight. The quantification of total anthocyanin content in the concentrated extract was measured by adding ten ml of pH buffers (either pH 1.0 and 4.5) to the vacuumed strawberry samples (step 2) and then incubated for 20 min at room temperature and centrifuged at 4 °C, 7000 rpm for 15 min. The supernatant was then removed, the absorbance was read at 520 and 700 nm, and anthocyanin was calculated by using Formula (2).

The tested solvents had the same dilution factor (solvent: sample ratio of 20:1; v/m). All the experiments had ten replicates, and the experiments were repeated twice. The average data were used to calculate anthocyanin concentration. The extracts were kept at 4 °C for 48 h before scanning for transmittance, color parameters, and profiling by UHPLC. After 48 h storage, samples extracted with acetone lost their color, and therefore, were not analyzed by UHPLC for anthocyanin profiling.

### 4.3. Transmittance Spectrum and Color Parameters

The extracts were stored at 4 °C for 48 h before being analyzed by Spectrophotometer CM-5 to measure their transmittance and the color parameters. The parameters of color were measured by tristimulus calorimetry [24] using a Konica Minolta CM-5 spectrophotometer (Minolta Co., Osaka, Japan), measuring wavelengths from 400–700 nm of liquid extracts with 10 nm intervals. Standard illuminant C was used as a reference. The samples were placed in 15 mL cuvettes with 1 cm path-length optical glass cells. CIE L*, a*, and b* values were measured in triplicate in the total transmission mode, using illuminant C and 2 observer angles and then calculating the average standard deviation. Chroma and Hue were calculated using Formulas (1) and (2):(3)$$\mathrm{Chroma}={\left(a^{2}+{\;b}^{2}\right)}^{\frac{1}{2}}$$
(4)$$\mathrm{Hue}=\tan^{-1}\left(\frac{b}{a}\right)$$

### 4.4. UHPLC Profiling of Individual Anthocyanins

Total anthocyanins extracted from fresh strawberry samples by different solvents were profiled by Ultra High Performance Liquid Chromatography (UHPLC). The extracts were stored at 4 °C for 48 h before being analyzed. All extracted samples were centrifuged again at 12,000 rpm and 4 °C for 10 min, and the supernatants were transferred to vials for LC-MS analysis. The separation and identification of individual anthocyanins in each extraction solvent was done using Vanquish UHPLC combined with Q Exactive MS (Thermo) and screened with ESI-MS. The LC system comprised an ACQUITY UPLC HSS T3 (100 × 2.1 mm × 1.8 μm) with Vanquish UHPLC. The mobile phase was composed of solvent A (0.1% formic acid water) and solvent B (0.1% formic acid acetonitrile) with gradient elution (0–2.0 min, 95% A; 2.0–15.0 min, 95–70% A; 15.0–15.1 min, 70–5% A; 15.1–20 min, 5% A; 20–20.1 min, 5–95% A; 20.1–26 min, 95% A). The flow rate of the mobile phase was 0.3 mL·min^−1^. The column temperature was maintained at 40 °C, and the sample manager temperature was set at 4 °C. Mass spectrometry parameters in ESI+ mode were listed as follows: Heater Temp 350 °C; Sheath Gas Flow rate, 40 arb; Aux Gas Flow Rate, 10 arb; Sweep Gas Flow Rate, 0 arb; spray voltage, 3.0 KV; Capillary Temp, 320 °C; and the collision energy at 20, 40, 60 eV.

The anthocyanins in the sample were identified by their retention times using standards for 103 anthocyanins (including but not limited to cyanidin-3-glucoside (C3G), malvidin-3-glucoside (M3G), pelargonidin-3-glucoside (P3G), pelargonidin-3-5-diglucoside (P3-5DG), and delphinidin-3-glucoside-chloride, all from Cerilliant, Round Rock, TX, USA).

### 4.5. Experimental Design and Statistical Analysis

All the experiments had ten replicates and were repeated three times, and the data were analyzed by using Statistical Analysis Software (SAS version 9.4; [25]). The experimental design was a completely randomized design.

## 5. Conclusions

Although new technologies (i.e., ultrasound and microwave-assisted extractions) have been developed to increase extraction yield and improve the stability of extracts, they need sophisticated instrumentation that increases the cost of anthocyanin production. According to Tena & Asuero [38], conventional extraction methods (maceration with solvents) are still the most widely used in the industry, particularly in natural dye industries, likely because these extraction methods have low instrumentation costs.

Several factors (i.e., sample type, pH, temperature, solvent type, and the solvent: sample ratio) affect anthocyanin yield, color parameters, and profile. Therefore, in this experiment, we kept the sample type, temperature, and solvent: sample ratio constant and compared different organic and water-based solvents for fresh strawberry puree. The first objective was to study the anthocyanin yield and color of the extracts from different solvents using the CIELAB parameters. The second objective was to study the anthocyanin profile of the extracted strawberries.

The solvent type changed the anthocyanin yield. The color parameters helped to identify any anthocyanin degradation over the 48 h of storage at 4 °C. When methanol [6] and methanol: water [7,9] solvents were used, anthocyanin yield was high, and the anthocyanins were stable for at least 48 h and even longer (data not presented) at 4 °C. The chloroform: methanol and ethanol was efficient in extracting anthocyanins, but anthocyanins degraded partially over the 48 h of storage at 4 °C. Water, water-based solvents (pH buffers), and acetone did not extract a high yield of anthocyanins. In the case of acetone, the extract degraded altogether over 48 h of storage.

The results of color parameter assessments suggested that an a* value lower than 30, L* values higher than 85, a hue angle more than 40, and a chroma less than 30 indicated some degree of color degradation in strawberry anthocyanins. Therefore, the methanol [6] and methanol: water [7,9] were the best solvents for anthocyanin assessment and their L*, a*, hue, and chroma parameters were similar. The second-best solvent was comprised by the pH differential buffers [11]. Other solvents such as ethanol [12], chloroform: methanol [6], water [12], and combined solvents [16] extracted considerable amounts of anthocyanins; however, they showed some degree of color degradation by measuring the color parameters. Acetone did not yield a stable extract and degraded over 48 h of the storage at 4 °C.

The extraction solvent determined the main anthocyanin of the anthocyanins profile. Pelargonidin was the major anthocyanin in chloroform: methanol solvent with a pH of 3.0, while delphinidin was dominant in all other solvents with pH valueS less than 2.0. The higher concentration of pelargonidin was responsible for the distinct color of the chloroform: methanol extract with lower a* and higher L* values compared to the methanol extract.

The data suggest using methanol and methanol: water as the solvents for extracting anthocyanins for comparison studies of different strawberry varieties. Additionally, measuring color parameters and anthocyanin profile will help to relate color to pigment and identify any degradation, as the human eye lacks linear sensitivity and cannot directly relate visible color to pigment concentration.

## Figures and Tables

**Figure 1 plants-12-01833-f001:**
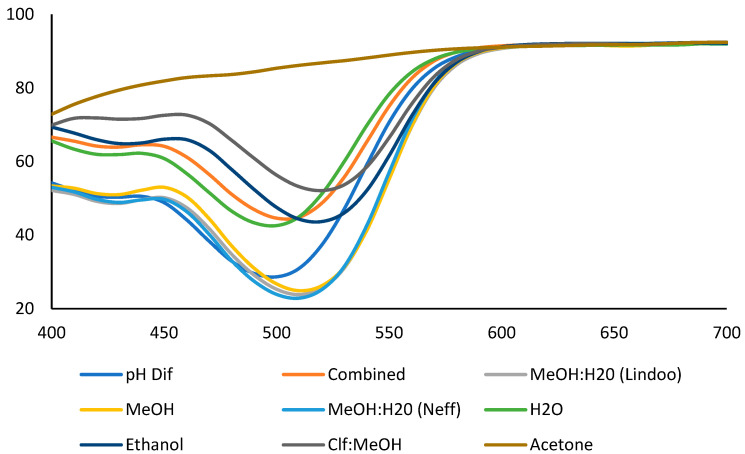
The transmittance spectrum of strawberry puree extracted by different extraction solvents after 48 h at 4 °C by Spectrophotometer CM-5. The solvents were (1) methanol (MeOH, Solovchenko et al. [6]), (2) methanol: water (MeOH:H_2_O, Neff and Chory [9]), (3) methanol: water (MeOH: H_2_O, Lindoo and Caldwell [7]), (4) ethanol (Karaaslan & Yaman 2017 [12]), (5) chloroform: methanol (Clf:MeOH, Solovchenko et al. [6]), (6) acetone (Garcia-Viguera et al. [17]), (7) water (Karaaslan & Yaman [12]), (8) pH differential buffers (Lee et al. [11]), (9) combined solvents (methanol: water and followed by pH differential; Gauch et al. [16]). The solvents were acidified by HCl (0.1%), except in Lindoo and Caldwell (1% HCl) and pH differential and combined solvents, where HCl was added to reach a certain pH (1.0 and 4.5). The extracts were stored at 4 °C for 48 h before measuring their transmittance.

**Figure 2 plants-12-01833-f002:**
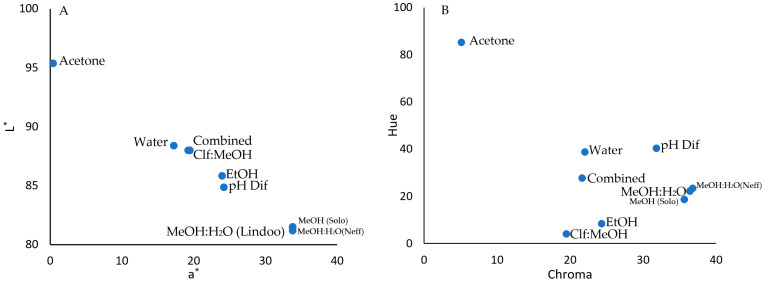
The color parameters (L*, a*, hue and chroma) of anthocyanins extracted from strawberry puree using nine different solvents (or their mixture) by CM-5 (Konica-Minolta). (**A**) L* and a* color parameters, (**B**) hue and chroma. The solvents were (1) methanol (MeOH, Solovchenko et al. [6]), (2) methanol: water (MeOH:H2O, Neff and Chory [9]), (3) methanol: water (MeOH: H2O, Lindoo and Caldwell [7]), (4) ethanol (EtOH, Karaaslan & Yaman, 2017 [12]), (5) chloroform: methanol (Clf:MeOH, Solovchenko et al. [6]), (6) acetone (Garcia-Viguera et al. [17]), (7) water (Karaaslan & Yaman [12]), (8) pH differential buffers (Lee et al. [11]), (9) combined solvents (methanol: water and followed by pH differential; Gauch et al. [16]). The solvents were acidified by HCl (0.1%), except in Lindoo and Caldwell (1% HCl) and pH differential and combined solvents, where HCl was added to reach a certain pH (1.0 and 4.5). The extracts were stored at 4 °C for 48 h before measuring their color parameters. Different letters for each parameter indicate significant differences at *p* ≤ 0.01.

**Table 1 plants-12-01833-t001:** Total Anthocyanin content of strawberries measured by different solvents and their combinations using spectrophotometer Genesys 150 UV-Vis [23].

Solvent Tested	Anthocyanin Concentration A/gFW
Chloroform: methanol (Solovchenko et al. [6])	11.9 ^a^
Ethanol (Karaaslan and Yaman [12])	11.0 ^b^
Methanol (Solovchenko et al. [6])	9.9 ^c^
Methanol: water (Neff and Chory [9])	9.4 ^c^
Acetone (Garcia-Viguera et al. [17])	8.8 ^d^
pH differential (Lee et al. [11])	7.5 ^e^
Methanol: water (Lindoo and Caldwell [7])	6.8 ^f^
Combined solvents (Gauch et al. [16])	4.9 ^g^
Water (Karaaslan and Yaman [12])	3.5 ^g^
LSD	0.56

Notes: The solvents were (1) methanol (Solovchenko et al. [6]), (2) methanol: water (Neff and Chory [9]), (3) methanol: water (Lindoo and Caldwell [7]), (4) ethanol (Karaaslan & Yaman [12]), (5) chloroform: methanol (Solovchenko et al. [6]), (6) acetone (Garcia-Viguera et al. [17]), (7) water (Karaaslan & Yaman [12]), (8) pH differential buffers (Lee et al. [11]), (9) combined solvents (methanol: water followed by pH differential; Gauch et al. [16]). The solvents were acidified by HCl (0.1%), except in Lindoo and Caldwell (1% HCl) and pH differential and combined solvents, where HCl was added to reach a certain pH (1.0 and 4.5). The same dilution factors were included in the formula for all solvents (20:1 v/m solvent: sample). Different letters indicate significant differences at *p* ≤ 0.01.

**Table 2 plants-12-01833-t002:** The color of anthocyanins extracted from strawberry puree using nine different solvents (or their combinations) and their measured color parameters (L*, a*, b*, hue and chroma) by CM-5 (Konica-Minolta).

Solvent	Color Parameters		pH	Color
Methanol: water (Neff and Chory [9])	L* 81.41 a* 33.89 b* 14.63 H 23.35 C 36.91	f a b d a	1.73	
Methanol: water (Lindoo and Caldwell [7])	L* 81.17 a* 33.84 b* 13.57 H 21.85 C 36.46	f a c d a	1.20	
Methanol (Solovchenko et al. [6])	L* 81.47 a* 33.84 b* 11.32 H 18.50 C 35.68	f a e e a	2.21	
pH differential (Lee et al. [11])	L* 84.84 a* 24.32 b* 20.56 H 40.21 C 31.84	e b a b b	1.67	
Ethanol (Karaaslan and Yaman [12])	L* 85.82 a* 24.07 b* 3.55 H 8.38 C 24.32	d b h f c	2.4	
Chloroform: methanol (Solovchenko et al. [6])	L* 87.959 a* 19.48 b* 1.34 H 3.93 C 19.53	c c i g e	3.02	
Combined solvents (Gauch et al. [16])	L* 88.0 a* 19.22 b* 10.04 H 27.58 C 21.68	c c f c d	1.63	
Water (Karaaslan and Yaman [12])	L* 88.37 a* 17.25 b* 13.77 H 38.61 C 22.07	c d c b d	1.95	
Acetone (Garcia-Viguera et al. [17])	L* 95.37 a* 0.45 b* 5.12 H 85.03 C 5.13	a f g a g	0.008	

Notes: The solvents were (1) methanol (Solovchenko et al. [6]), (2) methanol: water (Neff and Chory [9]), (3) methanol: water (Lindoo and Caldwell [7]), (4) ethanol (Karaaslan & Yaman [12]), (5) chloro-form: methanol (Solovchenko et al. [6]), (6) acetone (Garcia-Viguera et al. [17]), (7) water (Karaaslan & Yaman [12]), (8) pH differential buffers (Lee et al. [11]), (9) combined solvents (methanol: water followed by pH differential; Gauch et al. [16]). The solvents were acidified by HCl (0.1%), except in Lindoo and Caldwell (1% HCl) and pH differential and combined solvents, where HCl was added to reach a certain pH (1.0 and 4.5). The extracts were stored at 4 °C for 48 h before measuring their color parameters. Different letters for each parameter indicate significant differences at *p* ≤ 0.01 [25].

**Table 3 plants-12-01833-t003:** The profile of main anthocyanidins/anthocyanins (>10% of the total area under the peaks) extracted by different solvents from freshly-pureed strawberries and identified by ultra-high liquid chromatography-electrospray ionization tandem mass spectrometry (UHPLC-ESI-MS). The compound name, formula, retention time (RT), detected charge/mass ratio (*m*/*z*), the area under the peak, and the percentage of total area (%) are presented.

Solvent	Compound Name Formula	RT (min)	*m*/*z* Detect	*m*/*z* Adduct	Area	MS/MS Fragment	%
Chloroform: methanol (Solovchenko et al. [6])	Pelargonidin C_15_H_11_O_5_	10.04	271.0593	271.0601	1.62 × 10^6^	-	20.40
	Cyanidin C_15_H_11_O_6_	16.82	287.0536	287.0550	9.84 × 10^5^	287.06	12.36
	Petunidin C_16_H_13_O_7_	16.84	317.0651	317.0656	1.12 × 10^6^	317.07	14.09
Methanol (Solovchenko et al. [6])	Delphinidin C_15_H_11_O_7_	12.98	303.0491	303.0499	1.42 × 10^7^	303.05	87.09
Methanol: water (Lindoo and Caldwell [7])	Delphinidin C_15_H_11_O_7_	12.99	303.0491	303.0499	6.86 × 10^6^	303.05	50.86
pH differential (Lee et al. [11])	Delphinidin C_15_H_11_O_7_	12.99	303.0491	303.0499	8.97 × 10^6^	303.05	79.53
Methanol: water (Neff and Chory [9])	Delphinidin C_15_H_11_O_7_	12.97	303.0489	303.0499	1.50 × 10^7^	303.05	86.22
Combined solvents (Gauch et al. [16])	Delphinidin C_15_H_11_O_7_	12.98	303.0489	303.0499	2.47 × 10^6^	303.05	39.97
	Delphinidin-3,5-O-diglucoside C_27_H_31_O_17_	1.10	627.1594	627.1556	2.84 × 10^6^	465.10, 303.05	45.93
Water (Karaaslan and Yaman [12])	Delphinidin C_15_H_11_O_7_	12.89	303.0489	303.0499	2.74 × 10^6^	303.05	39.79
	Delphinidin-3,5-O-diglucoside C_27_H_31_O_17_	1.10	627.1594	627.1556	2.48 × 10^6^	465.10, 303.05	45.39
Ethanol (Karaaslan and Yaman [12])	Delphinidin C_15_H_11_O_7_	12.98	303.0493	303.0499	1.08 × 10^6^	303.05	45.84
	Pelargonidin C_15_H_11_O_5_	10.01	271.0600	271.0601	4.26 × 10^5^	-	18.14
	Cyanidin C_15_H_11_O_6_	16.82	287.0541	287.0550	4.06 × 10^5^	287.06	17.27

Notes: ‘-’ means UHPLC-ESI-MS did not detect the compound. Acetone-extracted anthocyanins changed color after 48h of storage at 4 °C and were not analyzed. The solvents were (1) Chloroform:methanol (Solovchenko et al. [6]), (2) methanol (Solovchenko et al. [6]), (3) methanol: water (Lindoo and Caldwell [7]), (4) pH differential buffers (Lee et al. [11]), (5) methanol: water (Neff and Chory [9]), (6) combined solvents (methanol: water followed by pH differential; Gauch et al. [16]), (7) water (Karaaslan & Yaman [12]), (8) ethanol (Karaaslan & Yaman [12]), The solvents were acidified by HCl (0.1%), except in Lindoo and Caldwell (1% HCl) and pH differential and combined solvents, where HCl was added to reach a certain pH (1.0 and 4.5). The extracts were stored at 4 °C for 48h before analyzing for anthocyanin profile.

## Data Availability

Data is contained within the article and will be provided on request.

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
