# Peer review of "Extraction Solvents Affect Anthocyanin Yield, Color, and Profile of Strawberries"

_plants, 2023, doi:10.3390/plants12091833_

Round 1
Reviewer 1 Report (Previous Reviewer 1)
The authors responded positively to all my suggestions.
Author Response
Dear Reviewer, Thank you very much for the comments and suggestions. We will improve the area mentioned in your comments.
Authors
Reviewer 2 Report (Previous Reviewer 4)
I thank the authors for their work.
I hope that the authors will take into account my wishes from the previous review and improve the methodology of future work.
Author Response
Dear Reviewer,
Thank you for your comments and suggestions. We will consider them in future research.
Authors
Reviewer 3 Report (New Reviewer)
The manuscript “Extraction solvents affect anthocyanin yield, color, and profile of strawberries” is devoted to the investigation of the influence of the solvent on the anthocyanin profile and content in strawberry extracts. A total of nine organic and water-based solvents (methanol and chloroform: methanol, acetone, ethanol, water) were studied and their effect on color parameters (saturation, hue and color degradation of strawberry anthocyanins) was determined. The authors found that solvents had a significant effect on yields, color parameters, and antocyanin profile. They showed that the best solvents for evaluating and obtaining anthocyanins were methanol and methanol:water. Anthocyanin compositions in samples of fresh strawberry extracts were determined by UHPLC-ESI-MS.
The authors did extensive job. However, such solvents as chloroform and methanol are toxic and can not be used in pharmaceutical or food industry. The authors should provide some examples of the fields, where the obtained results may possibly be used.
It is not clear where the authors got the standard compounds for UHPLC analysis.
In Conclusion section (lines 540-541) the authors wrote “Pelargonidin was the major anthocyanin in chloroform: methanol solvent with a pH of 3.0, while delphinidin was dominant in all other solvents with less than 2.0.” Did the authors mean “all other solvents with pH value less than 2.0.”? If so, please correct.
Author Response
1-A paragraph has been added to the conclusion to justify using the conventional extraction methods and their application, such as in the natural dye industry.
2-The source of the standards has been added to the Material and Methods.
3-Thank you for the comment. The correction has been made.
Reviewer 4 Report (New Reviewer)
I see nothing new in your work for me the article should not be accepted in this journal
Author Response
Dear Reviewer, Thank you very much for your comment. However, other reviewers have seen value in the manuscript and have not rejected the manuscript. Also, a different set of data from these experiments have been published, also cited in this manuscript, and has received a certificate as one of the most downloaded papers from Wiley, as seen below.
The main point of the paper is to emphasize the idea that different solvents extract different anthocyanin profiles, which will affect the color and stability of the extracts. Also, the color parameters are good indicators of anthocyanin degradation which can not be distinguished by eyes. Therefore, we think there is value in this set of experiments and the prepared manuscript.
Round 2
Reviewer 4 Report (New Reviewer)
Good luck
This manuscript is a resubmission of an earlier submission. The following is a list of the peer review reports and author responses from that submission.
Round 1
Reviewer 1 Report
All my suggestion are given in the form of comments in the attached pdf document.

Reviewer 2 Report
This paper describes how extraction solvents affect anthocyanin yield, color and profile of strawberries. Having re-read the paper, I consider that the paper has the same shortcomings as the previous paper presented by the authors. In my experience, in addition to the fact that the paper is not well designed, it has huge conceptual errors, such as the following:
- In order to compare the amount of anthocyanins extracted, an evaluation of the optimal extraction conditions should be carried out beforehand. The authors only evaluate the extraction solvents, but do not consider the interactions of the solvent with the rest of the variables, which are fundamental.
- The authors only use solid-liquid extraction as an extraction method, which is one of the least effective methods for extracting compounds. Currently there are highly efficient methods such as UAE, MAE, PLE that are able to extract a higher amount of anthocyanins in less time. The authors claim that UAE extraction does not use solvents, which is totally false.
- Measuring the level of anthocyanins by the colour of the extract, having extracts at different pH and different solvents does not provide any information. The colour of anthocyanins is known to be highly dependent on these two factors.
- The authors talk about the applicability of these extracts in food, but they use highly toxic solvents such as methanol or chloroform.
- I have serious doubts about the quantification of anthocyanins by UHPLC, from the point of view that they quantify mainly anthocyanidins rather than anthocyanins.
For all these main reasons, I consider that the work cannot be accepted for publication in plants under any circumstances.
Reviewer 3 Report
no further comments
Reviewer 4 Report
The manuscript “Extraction solvents affect anthocyanin yield, color, and profile of strawberries" is devoted to studying of anthocyanin extraction from strawberries. Methanol-water, ethanol, acetone, water and chloroform were used for extraction. Main anthocyanin profile was obtained using UHPLC-ESI-MS. Chroma and Hue parameters were calculated. This work has some significance from the point of view of extraction techniques and evaluation of the stability of the extracts obtained. In general, although the work has been done quite logically, its significance does not seem to be too high. Perhaps if the authors had offered their more optimal conditions or extraction methods, the significance of the work would have been higher.
However, it should be noted that the manuscript is written in a good language, has a good structure and a clear representation of the data. I think, this manuscript can be published in the Plants journal after minor revision after taking into account comments given below:
1. Chemical formulas should be written using subscripts.
2. Table 1: double of “Table 1”, letters a-g should be superscripts.
3. Table 2: the Color looks a little bit strange, because the color display may differ depending on the matrix of monitors and/or the quality of printers. Thus, it is not possible to use this data. It is better to specify the absorption maxima.
4. Check text style in all text, please.